# T Cell Immunity in Human Papillomavirus-Related Cutaneous Squamous Cell Carcinoma—A Systematic Review

**DOI:** 10.3390/diagnostics14050473

**Published:** 2024-02-21

**Authors:** Shi Huan Tay, Choon Chiat Oh

**Affiliations:** 1Duke-NUS Medical School, Singapore 169857, Singapore; tay_shi_huan@u.duke.nus.edu; 2Department of Dermatology, Singapore General Hospital, Singapore 169608, Singapore

**Keywords:** cutaneous squamous cell carcinoma, human papillomavirus, T cell, immunosuppression, animal model

## Abstract

Cutaneous squamous cell carcinoma (cSCC) is an invasive malignancy that disproportionately afflicts immunosuppressed individuals. The close associations of cSCC with immunosuppression and human papillomavirus (HPV) infection beget the question of how these three entities are intertwined in carcinogenesis. By exploring the role of T cell immunity in HPV-related cSCC based on the existing literature, we found that the loss of T cell immunity in the background of β-HPV infection promotes cSCC initiation following exposure to environmental carcinogens or chronic trauma. This highlights the potential of developing T-cell centred therapeutic and preventive strategies for populations with increased cSCC risk.

## 1. Introduction

Cutaneous squamous cell carcinoma (cSCC) is an invasive malignancy that arises from keratinocytes. Its rising global incidence and its severe burden on immunosuppressed individuals put forth an urgent need to develop novel approaches for preventing and treating cSCC [1,2,3,4].

The human papillomavirus (HPV) has been implicated as a risk factor for cSCC. There is an increased HPV prevalence (in particular β-HPVs) in cSCC compared to normal skin [5,6]. However, the causal relationship between HPV and cSCC remains controversial. HPV may exhibit direct oncogenic effects, but it may also act as a co-carcinogen with other risk factors (e.g., UVB radiation) to amplify the risk of developing cSCC [6,7,8,9].

Immunosuppression profoundly elevates the risk of cancers associated with viral infection, including cSCC. Immunosuppressed patients have up to 100-fold higher cSCC rates compared with the general population [10,11,12,13]. Antiviral immunity is chiefly regulated by the adaptive immune system, where T cells orchestrate effective long-lived responses. Immunosuppression profoundly diminishes T cell function, metabolism, and proliferation. This results in compromised protection against HPV proliferation in the skin, which likely contributes to carcinogenesis.

In this study, we aimed to clarify the role of T cell immunity in HPV-related cSCC based on the current literature. We identified potential causal relationships among T cell immunity, HPV, and cSCC, which may guide future preventive and therapeutic approaches, particularly in high-risk populations. 

## 2. Materials and Methods

A systematic search of research databases (PubMed, Embase, Scopus, Cochrane Library) was performed on 27 August 2023 in accordance with PRISMA guidelines (Figure 1; PROSPERO registry number CRD42023470491). Article screening and data extraction were performed in duplicate. Full-text studies (in vitro, in vivo, ex vivo, clinical) published in English that investigated T cell immunity in HPV-related cSCC were included.

The retrieved sources were screened by two independent authors (SHT and CCO) using titles and abstracts for inclusion. In situations where article suitability was uncertain, full text assessment was conducted, and discrepancies were resolved by a vote of consensus. Articles were selected based on the following inclusion criteria: (1) written in the English language, (2) using validated in vitro and in vivo models of HPV-related cSCC, ex vivo studies on patients with HPV-related cSCC, or randomised control trials (RCTs) on patients with HPV-related cSCC, and (3) having an emphasis on T cell immunity. Articles were excluded for the following reasons: (1) not reporting original data, (2) not focusing on HPV-related cSCC, (3) not focusing on T cell immunity, (4) observational clinical studies, and (5) lacking available full text. 

## 3. Results

Our literature search enabled us to retrieve 706 articles, from which 8 were included for the final qualitative analysis (Figure 1).

### 3.1. T Cell Immunity in HPV-Related cSCC Carcinogenesis

A total of six articles discussed T cell perturbations in HPV-related cSCC carcinogenesis (Table 1 and Table 2). All articles employed cSCC mouse models and one of the six reported additional data from human cSCC samples. For the cSCC mouse models, five of the six studies focused on β-HPVs—HPV8 mice were used in two of the six studies and *Mus musculus* papillomavirus 1 (MmuPV1)-colonised mice in three of the six studies. Only one of the six studies investigated α-HPVs with HPV16 mice. For carcinogenesis protocols, four of the six studies utilised spontaneous tumorigenesis, four of the six looked into ultraviolet B (UVB) irradiation, while one of the six investigated 7,12-Dimethylbenz(a)anthracene-12-O-tetradecanoylphorbol-13-acetate (DMBA-TPA) chemical carcinogenesis.

#### 3.1.1. T Cell Immunity in α-HPV-Related Epithelial Carcinogenesis

De Visser et al. (2005) hypothesised that an activated adaptive immunity promotes chronic inflammation in premalignant skin, thereby facilitating de novo epithelial carcinogenesis (Table 1) [14]. To address this, de Visser et al. used a transgenic mouse model of multistage epithelial carcinogenesis that expresses early region genes of HPV16 under the control of the human keratin 14 (K14) promoter/enhancer and is Recombination–Activating Gene-1 homozygous null (*Rag1*^−/−^) [15,16]. HPV16 is one of the most common high-risk α-HPVs, responsible for most HPV-related anogenital and head and neck cancers [17]. *Rag1*^−/−^ mice are deficient in mature B and T lymphocytes [16].

HPV16/*Rag1*^−/−^ mice had reduced infiltration of innate immune cells and minimal inflammation in premalignant skin, which was associated with a decreased cSCC incidence. However, the lack of mature CD4^+^ and/or CD8^+^ T lymphocytes alone (using *CD4*^−/−^:K14-HPV16, *CD8*^−/−^:K14-HPV16, *CD4*^−/−^*CD8*^−/−^:K14-HPV16 mice) did not replicate the clinical phenotype (Table 1). On the contrary, the adoptive transfer of B lymphocytes and serum transfer from HPV16 mice into HPV16/*Rag1*^−/−^ mice restored the characteristic chronic inflammation in premalignant skin and reinstated processes that are necessary for malignant progression. This study thus suggests a limited role for T cells in inflammation-associated α-HPV-driven, de novo epithelial carcinogenesis.

#### 3.1.2. T Cell Immunity in β-HPV-Related Epithelial Carcinogenesis

Of the five major HPV genera, β-HPVs are the most implicated genus in cSCC. β-HPVs are commensal viruses of the skin that are usually associated with asymptomatic infection in healthy individuals. However, several studies have reported increased β-HPV replication in the skin and greater β-HPV seropositivity in cSCC patients, which suggest a potential role for viral oncogenesis [18]. More importantly, the increased incidence of cSCC with concomitantly higher rates of β-HPV amongst solid organ transplants suggests a role for anti-β-HPV immunity in carcinogenesis [10,11,12,13]. 

Borgogna et al. (2023) and Antsiferova et al. (2017) utilised transgenic mice that express early region genes (encoding E1, E2, E4, E6, and E7) of HPV8, the prototypical β-HPV that is studied in HPV-related cSCC [19,20,21]. Borgogna et al. demonstrated accelerated papilloma development and greater accumulation of UVB-induced epidermal DNA damage in *Rag2*^−/−^:K14-HPV8 mice, which lack mature B and T lymphocytes (Table 2). They proposed that adaptive immune deficiency, such as that in solid organ transplant patients, sensitised β-HPV-infected skin to UVB-induced inflammation and promoted subsequent epithelial carcinogenesis. 

Antsiferova et al. [20] reported more epidermal CD4^+^ (including presumptive CD4^+^CD25^+^ regulatory T cells) and CD8^+^ T cells in tumorous skin of activin A-overexpressing mice compared to that of age-matched wild-type mice (Table 2). Activin A, a member of the TGF-β superfamily, is a growth and differentiation factor that promotes wound healing and skin morphogenesis [22]. It has been shown to be upregulated in skin wounds and human non-melanoma skin cancers (including cSCC) [22,23]. The authors also observed fewer epidermal gamma delta T cells for the same comparison (Table 2). However, CD4^+^ T cell depletion did not significantly reduce the tumour-promoting effect of activin A overexpression (Table 2). Hence, T cell perturbations alone appear to be insufficient for driving β-HPV-associated cSCC initiation, especially in the context of activin A overexpression. 

**Table 2 diagnostics-14-00473-t002:** Key T cell perturbations in β-HPV-related cSCC carcinogenesis (HPV8 mice).

Author and Year	Study Population	Key T Cell Perturbations
Borgogna et al. (2023) [19]	cSCC mouse models *Rag2*^−/−^:K14-HPV8 mice Spontaneous tumorigenesis, UVB	Genetic elimination of T and B cells increased spontaneous tumour incidence in *Rag2*^−/−^:K14-HPV8 mice (vs. *Rag2*^+/+^:K14-HPV8 mice; week 10: *p* < 0.05; Week 25: *p* < 0.0001) Genetic elimination of T and B cells increased percentage of spontaneously affected skin in *Rag2*^−/−^:K14-HPV8 mice (vs. *Rag2*^+/+^:K14-HPV8 mice; week 24: *p* < 0.0001) Genetic elimination of T and B cells increased percentage of affected skin following UVB irradiation in *Rag2*^−/−^:K14-HPV8 mice (vs. *Rag2*^+/+^:K14-HPV8 mice and non-transgenic control mice; week 30: both *p* < 0.01) Genetic elimination of T and B cells increased epidermal thickness following UVB irradiation in *Rag2*^−/−^:K14-HPV8 mice (vs. *Rag2*^+/+^:K14-HPV8 mice; *p* < 0.0001) Genetic elimination of T and B cells increased epidermal DNA damage following UVB irradiation in *Rag2*^−/−^:K14-HPV8 mice (vs. *Rag2*^+/+^:K14-HPV8 mice; γH2AX-positive nuclei: *p* < 0.001, 53BP1-positive foci: *p* < 0.001) Genetic elimination of T and B cells was associated with accumulation of epidermal DNA damage following UVB irradiation in *Rag2*^−/−^:K14-HPV8 mice (vs. *Rag2*^−/−^:K14-HPV8 mice without UVB irradiation; *p* < 0.0001)
Antsiferova et al. (2017) [20]	cSCC mouse models HPV8-Act, HPV8-wt mice ± CD4KO Spontaneous tumorigenesis	Accumulation of epidermal and dermal CD4^+^ and CD8^+^ T cells in tumour-laden ear skin of 10-week-old HPV8-Act mice (vs. age-matched wt mice, CD4: *p* < 0.0001, CD8: *p* = 0.0009; vs. age-matched HPV8-wt mice, CD4: *p* < 0.0001, CD8: *p* = 0.0022) Accumulation of epidermal CD4^+^CD25^+^ T cells in tumour-laden ear skin of 10-week-old HPV8-Act mice (vs. age-matched wt mice, *p* = 0.0004; vs. age-matched HPV8-wt mice, *p* = 0.0007) Large increase in tumour incidence in HPV8-Act-CD4KO mice vs. HPV8-wt-CD4KO (*p* < 0.0001) Slight but statistically insignificant increases in tumour incidence in HPV8-wt-CD4KO mice vs. HPV8-wt mice, and in HPV8-Act-CD4KO vs. HPV8-Act-wt (*p* = n.s.) Loss of epidermal gamma delta T cells in tumour-laden ear skin of 10-week-old HPV8-Act mice (vs. age-matched wt mice, *p* < 0.0001; vs. age-matched HPV8-wt mice, *p* = 0.0007)

Act, activin A-overexpressing; cSCC, cutaneous squamous cell carcinoma; HPV, human papillomavirus; KO, knockout; n.s., not significant; UVB, ultraviolet B; wt, wild-type.

Strickley et al. (2019), Johnson et al. (2022), and Dorfer et al. (2020) relied on another experimental model for studying commensal HPV interaction with human hosts: MmuPV1-colonised mice [24,25,26]. By doing so, these studies sought to interrogate T cell immunity through an infection-based system that models the natural history of HPV-related cSCC carcinogenesis [27]. 

Strickley et al. (2019) [24] observed that the adoptive transfer of T cells from MmuPV1-immune mice into wild-type FVB mice promoted wart rejection and protected against DMBA-TPA chemical carcinogenesis (Table 3). They also noticed an increased ratio of epidermal CD8^+^ tissue–resident memory T (T_RM_) cells to total T cells in the skin of MmuPV1-colonised mice compared to their sham-infected controls following chemical or UVB carcinogenesis (Table 3). Hence, the authors hypothesised that CD8^+^ T cells mediate anti-tumour immunity that is induced by MmuPV1 skin colonisation. They showed that CD8^+^ T cell depletion in MmuPV1-colonised mice increased the tumour incidence following chemical carcinogenesis (Table 3). Furthermore, they observed fewer CD8^+^ T cells and CD8^+^ T_RM_ cells alongside a higher β-HPV load in cSCC samples from immunosuppressed patients than in those from immunocompetent individuals (Table 3). β-HPV E7 peptides activated CD8^+^ T cells that were isolated from the normal facial skin of immunocompetent adults (Table 3). As such, the results showed that MmuPV1-immune mice were protected against epithelial carcinogenesis in a CD8^+^ T cell-dependent fashion. This finding suggests a role for commensal β-HPV-specific adaptive immunity in eliminating virus-positive malignant keratinocytes, thereby achieving anti-tumour protection.

Following up on Strickley et al. (2019) [24], Johnson et al. (2022) [25] investigated if a compromised T cell immunity could explain the increased β-HPV replication and seropositivity that is found in patients with an increased cSCC risk, such as those under immunosuppression. The authors demonstrated that CD8^+^ T cell depletion did increase the MmuPV-1 DNA levels in virus-colonised mouse skin and resulted in higher antibody titres to MmuPV1 E6, E7, and L1 antigens (Table 3). Interpreting both Strickley et al. (2019) and Johnson et al. (2022) in conjunction, it appears that the loss of T cell immunity against commensal β-HPVs confers an increased cSCC risk and higher viral load in immunosuppressed patients.

Dorfer et al. (2020) [26] also looked into how MmuPV1 infection can induce cSCC development in the context of immunosuppression. For this study, they treated mice with cyclosporine A (CsA), which inhibits calcineurin and preferentially suppresses T cell activation. The authors reported that MmuPV1 infection of back skin caused cSCC development in CsA-immunosuppressed mice but not in immunocompetent mice. Additionally, athymic NMRI-Foxn1^nu/nu^ mice developed secondary tumours after receiving intradermal administration of primary cSCC cells that were isolated from a cSCC of a MmuPV1-infected, CsA-immunosuppressed/UVB-treated mouse (Table 3). These primary cSCC cells were multiply passaged and lacked MmuPV1 DNA. Thus, this study concurs with the prior two articles that a deficient T cell immunity in the presence of β-HPV infection predisposes to cSCC initiation. It also implicates β-HPVs as non-essential in cSCC maintenance. 

### 3.2. T Cell Immunity in Potential Vaccination Strategies against β-HPV-Related cSCC Carcinogenesis

Marcuzzi et al. (2014) and Hufbauer et al. (2022) assessed potential vaccination strategies against β-HPV-related epithelial carcinogenesis by relying on the preclinical keratin-14 (K14)-HPV8 transgenic mouse model [28,29]. This model preferentially expresses all early genes (E1, E2, E4, E6, and E7) of HPV8 in the epidermis and developing hair follicles [21,30,31]. Viral gene expression is controlled by the human K14 promoter. At baseline, the viral antigens are synthesised at a subthreshold level that does not induce carcinogenesis, which is comparable to asymptomatic colonisation in immunocompetent individuals. Mechanical skin irritation from tattooing and/or tape-stripping induces epithelial carcinogenesis by activating high levels of HPV8 early gene expression.

Marcuzzi et al. (2014) [28] first showed that tattooing HPV8 E6 DNA onto the skin could prevent papilloma formation, which depends on anti-HPV8-E6-specific T cell immunity (Table 4). The HPV8 transgenic skin grafts of 6/15 tattooed (i.e., HPV-E6-immunised) mice did not develop papillomas after mechanical wounding. Following a HPV8 E6 epitope aa76-90 challenge and subsequent ELISpot assaying, splenocytes that were isolated from these six mice yielded a higher median number of spots (reflecting IFN-**γ**-producing cells per 100,000 splenocytes) than splenocytes from mice with papillomas (Table 4). Hence, a cytotoxic T cell response induced by skin tattooing of HPV E6 DNA may offer protection against HPV8-related epithelial carcinogenesis, albeit unreliably.

Hufbauer et al. (2022) [29] explored if innate immunity-driven in situ autovaccination against the patients’ own commensal β-HPV types in the skin could induce T cell immunity against β-HPV-related epithelial carcinogenesis in high-risk groups. Tattooing polyinosinic–polycytidylic acid (poly[I:C]) prevented tumour formation in eight out of eight treated mice. Poly(I:C) is a synthetic analogue of double-stranded RNA, a known ligand for the innate immune receptors TLR3 and MDA5 [32]. In poly(I:C)-treated non-tumorigenic sites, there were more activated CD4^+^ and CD8^+^ T cells than in untreated skin (Table 4). CD4^+^ T cell depletion and, to a smaller extent, CD8^+^ T cell depletion resulted in tumour formation in poly(I:C)-treated sites (Table 4). CD4^+^ T cell depletion also resulted in larger tumour sizes in poly(I:C)-treated sites compared to CD8^+^ T cell depletion (Table 4). As such, CD4^+^ T cells are likely the main effectors of poly(I:C)-mediated protection against HPV8-related epithelial carcinogenesis.

## 4. Discussion

The coexistence of impaired T cell immunity, β-HPV infection, and carcinogen exposure (such as UVB irradiation and DMBA) or chronic trauma promote cSCC initiation (Figure 2). An impaired T cell immunity exists in certain populations, such as organ transplant recipients on chronic immunosuppression, atypical epidermodysplasia verruciformis (EV), and EV-like phenotypes [10,11,12,13,33,34,35]. Consequently, these individuals possess markedly weakened anti-β-HPV defences, which are primarily orchestrated by T cells. The compromised β-HPV-specific T cell immunity reduces the clearance of β-HPVs and virus-positive malignant keratinocytes that have spawned following carcinogen exposure or chronic trauma, thereby potentiating cSCC initiation.

The host specificity of HPV has restricted the translatability of preclinical models when studying HPV-related cSCC [36]. An HPV transgenic mouse or animal papillomavirus-based infection model does not fully replicate the complex skin microbiome of human skin, is limited by inherent discrepancies in both innate and adaptive immunity, and may be affected by variations in experimental conditions. Nevertheless, the multitude of cSCC mouse model studies that were reviewed in this article have proved invaluable in revealing the anti-tumour effects of β-HPV-specific T cells in a tractable manner, which is otherwise not possible to conduct in human studies.

There is a significant lack of granularity regarding the specific T cell perturbations in HPV-related cSCC carcinogenesis. The included studies primarily relied on the enumeration of total T cell counts in affected skin and the depletion of total CD4^+^ and/or CD8^+^ T cells. Given that T cell diversity is wide-ranging, ranging from effector to regulatory, it is reasonable to hypothesise that specific T cell populations drive anti-tumour immunity. Addressing the β-HPV specificity of these populations is also pertinent, as the findings will provide crucial mechanistic evidence for whether the loss of β-HPV-specific T cell immunity or the de novo oncogenic effect of β-HPVs predominantly raises cSCC risk. Doing so will bridge key findings on T cell immunity in non-HPV-related cSCC models, where tumour-specific cytotoxic T cells (primarily Th1 and CD8^+^ T cells) inhibit UVB and/or chemical carcinogenesis, while tumour-infiltrating regulatory T cells (Tregs) likely suppress anti-tumour immunity [37].

Topical imiquimod has emerged as a potential treatment for actinic keratosis, the prototypical premalignant lesion of cSCC that often possesses high β-HPV loads [38]. Its promise has been highlighted in pre-invasive α-HPV-related neoplasms of the female lower tract, whereby imiquimod may be a valid, cost-effective first-line treatment to avoid surgical excision [39]. Imiquimod directly induces apoptosis of malignant keratinocytes and partially overcomes HPV E6/E7 activity to stimulate robust Th1-Th17 responses [39]. Thus, it would be interesting to explore if imiquimod is effective as a monotherapy or in combination with other modalities to impair HPV-related cSCC initiation by enhancing β-HPV-specific T cell immunity.

The advent of spatial omics technologies can address these issues by permitting in tumorous and non-tumorous skin the high-dimensional interrogation of immune perturbations extending beyond just T cells [40,41]. The in situ single cell-level profiling would greatly complement traditional reductionist approaches in resolving the complexities of the tumour microenvironment, by uncovering spatio-temporal relationships between T cells, malignant keratinocytes, and other contributors to carcinogenesis. Doing so will clarify the role of T cells and simultaneously assess other cellular players like macrophages in protecting against HPV-related cSCC. Another advantage of these technologies is their amenability to limited tissue samples, facilitating ex vivo human studies. Discoveries via these modalities can rapidly aid explorations and validation in animal models, thereby seeding the future for improved therapy and prevention in high-risk populations.

All in all, T cells are intimately involved in the defence against HPV-related cSCC (specifically β-HPV), as their deficiency potentiates carcinogenesis in high-risk populations. Studies integrating omics approaches and appropriate animal models are warranted to elucidate T cell-mediated immunosurveillance and inhibition of HPV-related cSCC initiation. In parallel, further characterisation of the skin virome in immunocompetent and immunosuppressed individuals will shed light on the immunogenicity of different β-HPV types and the viruses’ differential contributions to carcinogenesis. Future work can build upon these mechanistic studies to focus on protecting high-risk individuals with the prospects of T cell-centred vaccines against commonly occurring β-HPVs, β-HPV-specific T cell immunotherapy, and prognostication with β-HPV-specific T cells.

## Figures and Tables

**Figure 1 diagnostics-14-00473-f001:**
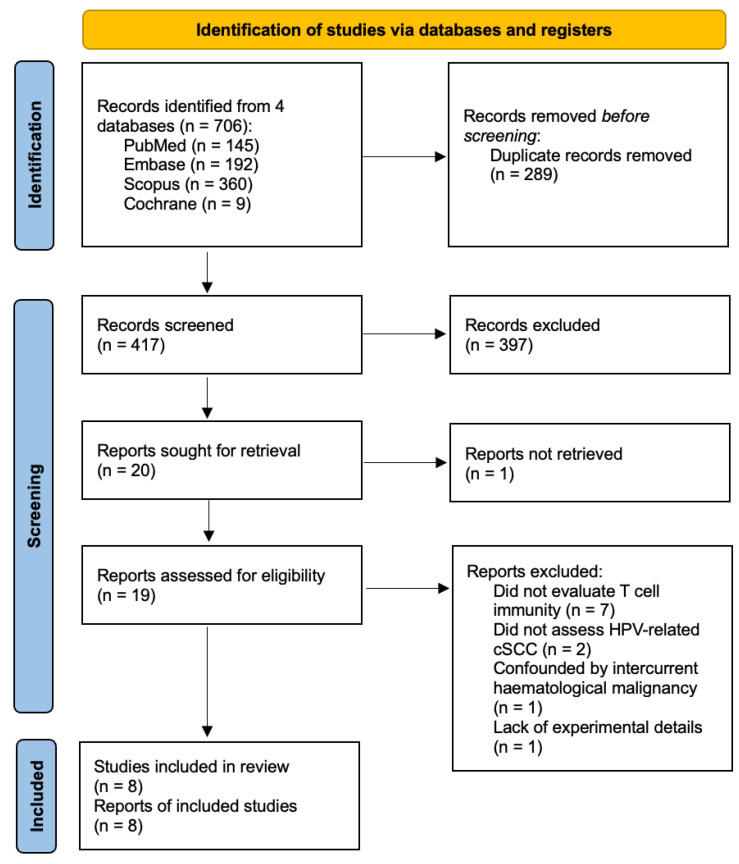
PRISMA flow diagram.

**Figure 2 diagnostics-14-00473-f002:**
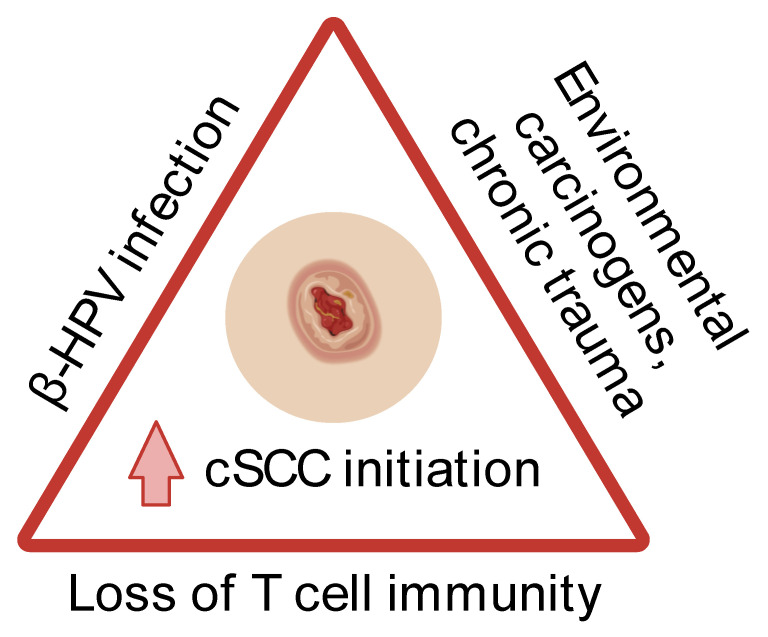
Loss of T cell immunity in the background of β-HPV infection promotes cSCC initiation following exposure to environmental carcinogens or chronic trauma. β-HPV, beta human papillomavirus; cSCC, cutaneous squamous cell carcinoma. Created with BioRender.com (accessed on 31 December 2023).

**Table 1 diagnostics-14-00473-t001:** Key T cell perturbations in α-HPV-related cSCC carcinogenesis.

Author and Year	Study Population	Key T Cell Perturbations
De Visser et al. (2005) [14]	cSCC mouse models *Rag1*^−/−^:K14-HPV16 mice, *CD4*^−/−^:K14-HPV16 mice, *CD8*^−/−^:K14-HPV16 mice, *CD4*^−/−^*CD8*^−/−^:K14-HPV16 mice Spontaneous tumorigenesis	Genetic elimination of CD4^+^ and/or CD8^+^ T cells did not reduce mast cell and granulocyte recruitment into premalignant skin (vs. K14-HPV16 mice, *p* = n.s.)

cSCC, cutaneous squamous cell carcinoma; HPV, human papillomavirus; K14, keratin 14; n.s., non-significant.

**Table 3 diagnostics-14-00473-t003:** Key T cell perturbations in β-HPV-related cSCC carcinogenesis (MmuPV1-colonised mice).

Author and Year	Study Population	Key T Cell Perturbations
Strickley et al. (2019) [24]	cSCC mouse models MmuPV1-colonised FVB mice, MmuPV1-colonised SKH-1 mice DMBA-TPA (FVB), DMBA-UVB (SKH-1) Human cSCC patients (immunosuppressed, immunocompetent)	Adoptive transfer of T cells from MmuPV1-immune mice into wild-type FVB mice with persistent warts reduced skin wart burden (vs. mice that received control T cells from spleen of uninfected wild-type FVB mice, *n* = 3 each group) Adoptive transfer of T cells from MmuPV1-immune mice into wild-type FVB mice promoted wart rejection and protected against DMBA-TPA chemical carcinogenesis (vs. mice that received control T cells from spleen of uninfected wild-type FVB mice, *n* = 3 each group) Increased ratio of epidermal CD8^+^ T_RM_ cells to total T cells in the skin of MmuPV1-colonised mice following DMBA-TPA chemical carcinogenesis (vs. sham-infected mice, *p* = 0.0287) and DMBA-UVB carcinogenesis (vs. sham-infected mice, *p* = 0.0054) More tumour-infiltrating CD8^+^ T cells in MmuPV1-colonised mice following DMBA-TPA chemical carcinogenesis (vs. sham-infected mice, *p* = 0.0208) CD8^+^ T cell depletion in MmuPV1-colonised mice increased tumour incidence following DMBA-TPA chemical carcinogenesis (vs. IgG-treated immunocompetent control mice, *p* = 0.0009) Fewer tumour- and skin-infiltrating CD8^+^ T cells and CD8^+^ T_RM_ cells in cSCC of immunosuppressed patients (vs. cSCC of immunocompetent patients; tumour-infiltrating CD8^+^ T and CD8^+^ T_RM_: both *p* < 0.0001; skin CD8^+^ T: *p* = 0.0001; skin CD8^+^ T_RM_: *p* = 0.0009) CD8^+^ T cells from normal facial skin of immunocompetent adults activated by β-HPV E7 peptides (vs. negative control; CD69^+^: *p* < 0.01, CD137^+^CD69^+^: *p* < 0.01), but not by high-risk α-HPV HPV16 E7 peptides (vs. negative control; CD69^+^: *p* = n.s., CD137^+^CD69^+^: *p* = n.s.)
Johnson et al. (2022) [25]	cSCC mouse model MmuPV1-colonised SKH-1 mice DMBA-UVB	CD8^+^ T cell depletion increased MmuPV1 DNA levels in virus-colonised mouse skin following DMBA-UVB carcinogenesis (vs. IgG-treated immunocompetent control mice; *p* = 0.0229) CD8^+^ T cell depletion resulted in higher antibody titres to MmuPV1 E6, E7, and L1 antigens following DMBA-UVB carcinogenesis (vs. IgG-treated immunocompetent control mice; E6: *p* = 0.0030, E7: *p* = 0.0220, L1: *p* = 0.0041)
Dorfer et al. (2020) [26]	cSCC mouse models MmuPV1-colonised FVB mice ± CsA immunosuppression NMRI-Foxn1^nu/nu^ mice Spontaneous tumorigenesis, UVB	MmuPV1 infection of back skin resulted in cSCC development in CsA-immunosuppressed mice (non-UVB-treated: *n* = 7/10; UVB-treated: *n* = 9/20), but not in immunocompetent mice (non-UVB-treated: *n* = 0/10; UVB-treated: *n* = 0/5) MmuPV1 infection increased mean CD4^+^ T cell numbers in back skin tissue of CsA-immunosuppressed/non-UVB-treated mice (vs. non-infected, equally treated controls; *p* < 0.05) Non-tumorous back skin in MmuPV1-infected, CsA-immunosuppressed/UVB-treated mice had higher CD8^+^ T cell numbers (vs. non-tumorous back skin in CsA-immunosuppressed/non-UVB-treated mice; *p* < 0.05) Higher FoxP3^+^ T cell numbers in tumorous back skin of MmuPV1-infected, CsA-immunosuppressed/non-UVB-treated mice (vs. non-tumorous back skin of same mice; *p* < 0.05) Intradermal administration of primary cSCC cells of passage 11 (MmuPV1 DNA undetectable) to athymic NMRI-Foxn1^nu/nu^ mice gave rise to secondary tumours at 30 days post-inoculation (*n* = 2)

CsA, cyclosporine A; cSCC, cutaneous squamous cell carcinoma; DMBA, 7,12-Dimethylbenz(a)anthracene; HPV, human papillomavirus; MmuPV1, *Mus musculus* papillomavirus 1; n.s., not significant; TPA, 12-O-tetradecanoylphorbol-13-acetate; T_RM_, tissue-resident memory T; UVB, ultraviolet B; wt, wild-type.

**Table 4 diagnostics-14-00473-t004:** Key T cell perturbations in potential vaccination strategies against HPV-related cSCC carcinogenesis.

Author and Year	Study Population	Vaccination Strategy	Key T Cell Perturbations
Marcuzzi et al. (2014) [28]	cSCC mouse model K14-HPV8-CER mice Mechanical wounding	HPV8 E6 DNA tattooing onto skin	Higher median number of spots reflecting IFN-**γ**-producing cells per 100,000 splenocytes (via IFN-**γ** ELISpot) following HPV8 E6 epitope aa76-90 challenge in splenocytes of DNA-immunised mice without papilloma (vs. with papilloma, *p* < 0.00001)
Hufbauer et al. (2022) [29]	cSCC mouse model K14-HPV8-CER mice Mechanical wounding	Poly(I:C) tattooing onto skin	More total and activated CD4^+^ T cells detected in poly(I:C)-treated non-tumorigenic sites (vs. untreated skin, total CD4: *p* < 0.001, activated CD4: *p* < 0.01) More activated CD8^+^ T cells detected in poly(I:C)-treated non-tumorigenic sites (vs. untreated skin, *p* < 0.01) CD4^+^ T cell depletion resulted in tumour formation in poly(I:C)-treated sites (*n* = 5/6 mice) CD8^+^ T cell depletion resulted in tumour formation in poly(I:C)-treated sites, but to a smaller extent (*n* = 2/6 mice) CD4^+^ T cell depletion resulted in larger tumour sizes in poly(I:C)-treated sites (vs. CD8^+^ T cell depletion)

aa, amino acid; CER, complete early genome region; cSCC, cutaneous squamous cell carcinoma; ELISpot, enzyme-linked immunosorbent spot; HPV, human papillomavirus; IFN, interferon; K14, keratin-14; poly(I:C), polyinosinic-polycytidylic acid.

## Data Availability

No new data were created or analysed in this study. Data sharing is not applicable to this article.

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
