# Peer review of "T Cell Immunity in Human Papillomavirus-Related Cutaneous Squamous Cell Carcinoma—A Systematic Review"

_diagnostics, 2024, doi:10.3390/diagnostics14050473_

Round 1

Reviewer 1 Report

Comments and Suggestions for Authors

This systematic review delves into the complex interplay between T cell immunity, human papillomavirus (HPV) infection, and the onset of cutaneous squamous cell carcinoma (cSCC), particularly in immunocompromised individuals. The authors highlight the disproportionate impact of cSCC on those with weakened immune systems and explore the pivotal role of T cell immunity in the context of β-HPV infections. Their investigation synthesizes findings from various studies, utilizing both animal models and clinical data, to shed light on the mechanisms by which T cell perturbations contribute to cSCC development.

The review meticulously outlines the methodology of literature selection based on PRISMA guidelines, ensuring a comprehensive and unbiased analysis of the available evidence. Through the inclusion of studies involving in vitro, in vivo, ex vivo, and clinical research, the authors provide a holistic view of the current understanding of T cell immunity's role in HPV-related cSCC.

The findings from the included studies suggest that the loss of T cell function, particularly in the context of β-HPV infections, can exacerbate the risk of cSCC by impairing the body's ability to control HPV proliferation and respond to environmental carcinogens. This is especially evident in immunosuppressed populations, such as organ transplant recipients, who exhibit a significantly higher incidence of cSCC. The review also discusses the potential for T-cell-centered therapeutic and preventive strategies, emphasizing the need for future research to further elucidate the specific mechanisms of T cell-mediated immunity in HPV-related cSCC.

Overall, this systematic review provides valuable insights into the interplay between HPV infection, T cell immunity, and cSCC, highlighting the importance of T cell function in preventing HPV-related carcinogenesis. It calls for further investigation into targeted immune-based therapies and the development of preventive measures for populations at increased risk of cSCC.

Introduction:

The introduction provides a comprehensive background on cutaneous squamous cell carcinoma (cSCC), emphasizing its rising incidence and the link with human papillomavirus (HPV) infection and immunosuppression. It successfully outlines the urgency for novel prevention and treatment strategies, referencing relevant studies [1-4] that justify the research focus. The cited references are relevant, laying a solid foundation for the study's rationale.

Cited References:

The cited references throughout the paper are pertinent and directly support the narrative on the role of T cell immunity in HPV-related cSCC. The references are up-to-date, covering essential background information, methodological approaches, and previous findings in the field, contributing to the paper's credibility and scholarly context.

Research Design:

The research design, employing a systematic review following PRISMA guidelines, is appropriate and rigorous for summarizing and analyzing existing literature on the topic. The inclusion and exclusion criteria for article selection are clearly defined, ensuring a focused and relevant review scope.

Methods:

The methods section is detailed, describing the systematic search strategy, databases searched, and the selection process, including the use of a PRISMA flow diagram. The dual-review approach for screening and data extraction enhances the reliability of the included studies.

Results:

The results are clearly presented, with the qualitative analysis of 8 included articles systematically organized into themes related to T cell immunity's role in HPV-related cSCC carcinogenesis and potential vaccination strategies. The use of tables to summarize key findings from the articles enhances clarity and allows for easy comparison.

Conclusions:

The conclusions are well-supported by the results, highlighting the critical role of T cell immunity in the context of HPV infection and cSCC. The paper effectively synthesizes the evidence from the reviewed articles, suggesting directions for future research and potential therapeutic strategies.

Conflict of Interest:

The authors have declared no conflict of interest, which is essential for maintaining the integrity and impartiality of the review.

Plagiarism:

There is no indication of plagiarism. The authors have synthesized information from various sources effectively, maintaining originality in their analysis and discussion.

Inappropriate Self-Citations:

There are no apparent inappropriate self-citations. The references cited are relevant to the study's scope and contribute meaningfully to the narrative.

Ethical Concerns:

No ethical concerns are evident in this study. The nature of a systematic review, primarily involving the synthesis of published literature, typically involves minimal ethical risk.

Comments on the Quality of English Language

Additional Comments:

The paper is well-structured and written, following the conventions of a systematic review. It provides a valuable synthesis of current knowledge on T cell immunity in HPV-related cSCC, identifying gaps and suggesting future research directions.

Author Response

Thank you for the valuable feedback. We deeply appreciate your detailed and positive comments on the robustness and clinical applicability of the study.

We have also added a paragraph discussing imiquimod's potential as a possible first-line, cost-effective therapy to impair HPV-related cSCC initiation (highlighted in red; Lines 233-241) as per another reviewer's input.

Reviewer 2 Report

Comments and Suggestions for Authors

Dear authors,

this is an interesting paper on the role of t cells immunity in the development of squamous cell carcinoma. It can be a relevant addition to the understanding oncogenic mechanism in HPV related carcinomas. 

Regarding methodology, the authors have used an adequate prisma guidelines so it is fine.

Conclusions are supported and consistent with evidences. Also, references are appropriate.

However I would suggest to improve the quality of the paper adding a paragraph about the emerging role of imiquimod in stimulating t cells and enhance immunity (cite 10.1002/jmv.29238). this would improve the completeness of the paper.

thank you for your precious work

Comments on the Quality of English Language

 minor

Author Response

Thank you for the valuable feedback. We agree that the emerging role of imiquimod in tackling HPV-related neoplasms should be highlighted in our review. Thus, we have added a paragraph discussing imiquimod's potential as a possible first-line, cost-effective therapy to impair HPV-related cSCC initiation (highlighted in red; Lines 233-241).

Reviewer 3 Report

Comments and Suggestions for Authors

Overall, this was a comprehensive review of T-cell immunity in squamous cell carcinoma associated with HPV. The authors did an excellent literature search.

Author Response

(The authors gave the same response as above.)

Round 2

Reviewer 2 Report

Comments and Suggestions for Authors

Thank you for your precious revision.

The paper deserves publication

Thank you